# Influence of Glucocorticoids on Cellular Senescence Hallmarks in Osteoarthritic Fibroblast-like Synoviocytes

**DOI:** 10.3390/jcm10225331

**Published:** 2021-11-16

**Authors:** Olivier Malaise, Geneviève Paulissen, Céline Deroyer, Federica Ciregia, Christophe Poulet, Sophie Neuville, Zelda Plener, Christophe Daniel, Philippe Gillet, Chantal Lechanteur, Jean-Marc Brondello, Dominique de Seny, Michel Malaise

**Affiliations:** 1Laboratory of Rheumatology, GIGA Research, CHU de Liège, University of Liège, 4000 Liège, Belgium; Genevieve.Paulissen@chuliege.be (G.P.); Celine.Deroyer@chuliege.be (C.D.); ciregia@gmail.com (F.C.); Christophe.Poulet@chuliege.be (C.P.); Sophie.Neuville@chuliege.be (S.N.); Zelda.Plener@chuliege.be (Z.P.); ddeseny@chuliege.be (D.d.S.); Michel.Malaise@chuliege.be (M.M.); 2Orthopedic Surgery Department, CHU de Liège, 4000 Liège, Belgium; Christophe.Daniel@chuliege.be (C.D.); Philippe.Gillet@chuliege.be (P.G.); 3Laboratory of Cell and Gene Therapy, Department of Hematology, CHU de Liège, 4000 Liège, Belgium; C.Lechanteur@chuliege.be; 4Institute for Regenerative Medicine and Biotherapy, Univ Montpellier, INSERM UMR1183, 34298 Montpellier, France; jean-marc.brondello@inserm.fr

**Keywords:** osteoarthritis, synovial membrane, clinical studies

## Abstract

Osteoarthritis (OA) is recognized as being a cellular senescence-linked disease. Intra-articular injections of glucocorticoids (GC) are frequently used in knee OA to treat synovial effusion but face controversies about toxicity. We investigated the influence of GC on cellular senescence hallmarks and senescence induction in fibroblast-like synoviocytes (FLS) from OA patients and mesenchymal stem cells (MSC). Methods: Cellular senescence was assessed via the proliferation rate, β-galactosidase staining, DNA damage and CKI expression (p21, p16^INK4A^). Experimental senescence was induced by irradiation. Results: The GC prednisolone did not induce an apparent senescence phenotype in FLS, with even higher proliferation, no accumulation of β-galactosidase-positive cells nor DNA damage and reduction in p21mRNA, only showing the enhancement of p16^INK4A^. Prednisolone did not modify experimental senescence induction in FLS, with no modulation of any senescence parameters. Moreover, prednisolone did not induce a senescence phenotype in MSC: despite high β-galactosidase-positive cells, no reduction in proliferation, no DNA damage and no CKI enhancement was observed. Conclusions: We provide reassuring in vitro data about the use of GC regarding cellular senescence involvement in OA: the GC prednisolone did not induce a senescent phenotype in OA FLS (the proliferation ratio was even higher) and in MSC and did not worsen cellular senescence establishment.

## 1. Introduction

Osteoarthritis (OA) is a consequence of the accumulation of age-induced damage in the joint, leading not only to cartilage degeneration, osteophytosis, or sub-chondral bone remodeling but also synovial hypertrophy or joint effusion. Progress has been made on physiopathology and OA is now considered to be a senescence-linked disease [1,2,3]. Cellular senescence is characterized by a robust cell cycle arrest with changes in cellular morphology and the expression of metabolic enzymatic activities, such as acidic senescence-associated β-galactosidase [4]. Several cyclin-dependent kinase inhibitors (CKI) are over-expressed, like p16^INK4A^ and p21 proteins, leading to cell proliferation arrest. The senescence program leads cells toward a senescence-associated secretory phenotype (SASP) secretion, which disturbs tissue homeostasis by producing metalloproteinases or inflammatory proteins, enhancing the spread of senescence. Depending on the studied cell types or their induction mode, several subsets of cellular senescence can co-exist in the organism [5]. 

The synovial membrane is no longer considered a bystander in OA: synovial hypertrophy is found in 1/3 of patients with OA [6] and this synovitis is associated with subsequent OA development [7]. Chronic low-grade inflammation is present in the synovial membrane; infiltration by immune cells [8] and the cross-talk between cartilage and synovial cells have a major role in OA pathogenesis [9]. In addition, we identified senescent cells in the synovium of OA mice [10] and Diekman et al. demonstrated that cartilage senescence could not be the only driver of the pathology [11].

There is still no validated drug to cure OA. Besides, pharmacological management is mainly restricted to painkillers, anti-inflammatory drugs and intra-articular glucocorticoid (GC) injections. GC injections are usually performed when effusion and synovitis occur, giving a short-term reduction in synovial inflammation. It is estimated that nearly 30% of new knee-OA patients have received at least one intra-articular injection [12]. Even if GC injections are part of the recent guidelines, they remain under debate. Indeed, a worsening of the disease is observed with iterative injections in humans [13]. Furthermore, in vitro GC are suspected as being deleterious for cartilage explant and chondrocytes [14,15] and the powerful GC dexamethasone is associated with higher senescence markers in chondrocytes [16] and tenocytes [17]. With regard (a) to the widespread use of intra-articular GC injection to face the joint effusion present in OA, (b) to the concerns about GC use in OA, and (c) to the emerging role of cellular senescence to explain OA onset, in this work we aim at determining the influence of a short course of stimulation with GC on the senescence parameters in fibroblast-like synoviocytes (FLS) from OA patients. Therefore, baseline senescence hallmarks and senescence induction have been analyzed under GC stimulation. 

## 2. Materials and Methods

### 2.1. Patient Recruitment

Regarding synovial biopsies, as used for immunohistochemistry (Figure 1), samples of synovial biopsies were retrospectively selected from our local cohort of OA patients: the synovial biopsies of OA patients (*n* = 16) were obtained by needle arthroscopy from affected knees. Samples were extracted and stocked as previously described by de Seny et al. [8]. Patients were mostly female, with a median age of 58 years (range 29–89) and a median BMI of 30 kg/m^2^ (range 18–42). 

Concerning the FLS used for in vitro experiments, the synovial membranes were obtained in collaboration with the orthopedic surgery department (CHU of Liege) from OA patients undergoing knee replacement surgery. Regarding patients included in experiments over 8 days (Figures 2 and 3, Appendix A) (*n* = 24), the median age was 66 years (range 51–81) and the median BMI was 31 kg/m^2^ (range 23–47). There were 16 females and 8 males. The median Kellgren and Lawrence grade was 4 (range 3–4). Regarding those included in experiments over 15 days (Appendix A) (*n* = 7), the median age was 60 years (range 55–74), the median BMI was 29 kg/m^2^ (23–36) and there were 2 females and 5 males. The median Kellgren and Lawrence grade was 4 (range 3–4). MSCs were obtained from healthy donors (bone-marrow aspiration) and were provided by the laboratory of cell and gene therapy (CHU of Liege) (Figure 4 and Appendix A). MSCs were produced in agreement with good manufacturing practices, as previously described [18]. Tissue and cell collections for this study were approved by the ethical committee of the CHU of Liege (B707201732662; ref: 2017/147) and the biobank research committee (BB190058) of the CHU of Liege. Both gave approval for studies collecting these tissues from patients of the CHU of Liege.

### 2.2. Immunohistochemistry (IHC)

For p16^INK4A^ IHC, CINtec Histology monoclonal mouse anti-human p16^INK4A^ (clone E6H4; Roche Tissue Diagnostics, Indianapolis, IN, USA) was used according to the manufacturer’s instructions. The slides were deparaffinized and then pretreated with the cell-conditioning media provided by the manufacturer for 48 min, with an OptiView DAB detection kit on Ventana BenchMark ULTRA autostainer. The Ventana predilute CINtec p16 Histology was used as the primary antibody (12 min, 36 °C). A counter-coloration with hematoxylin was performed over a period of 4 min. Staining was detected with a Nanozoomer Digital Pathology 2.0 HT scanner (Hamamatsu Photonics, Hamamatsu, Japan). The images derived from IHC were entirely quantified with the bioimage analysis software QuPath (version 0.2.3, https://QuPath.github.io/, accessed on 1 September 2021).

### 2.3. Cell Culture

Human FLS were isolated from an OA knee joint and cultured in Dulbecco’s modified Eagle’s medium (DMEM) supplemented with 10% fetal bovine serum (FBS), 1% l-glutamine (200 mM), 100 units/mL penicillin and 100 μg/mL streptomycin (BioWhittaker, Walkersville, MD, USA), as described previously [19]. MSCs were cultured in DMEM-low glucose with glutamax (Thermo Fisher Scientific, Waltham, MA, USA) supplemented with 10% FBS, 100 units/mL penicillin and 100 μg/mL streptomycin, as described by Lechanteur et al. [18]. Cells were maintained at 37 °C in a 5% CO_2_ atmosphere. In order to analyze the influence of GC, cells were plated at day 0 in 6- or 96-well plates and then stimulated with prednisolone (Sigma-Aldrich, Saint Louis, MO, USA) over a period of seven days (at days 1, 4, and 7) with three different doses (1, 5 or 10 µM). Cells were harvested on day 8 or 15. To induce senescence, cells were irradiated on day 1 with a unique dose of 10 Gray (Gy) using a Gammacell 40 Exactor (Nordion, Ottawa, ON, Canada). 

### 2.4. BrdU Cell Proliferation Assay

Cell proliferation was calculated using a BrdU Cell Proliferation ELISA Kit (Abcam, Cambridge, MA, USA), following the manufacturer’s instructions. On day 0, 5 × 10^3^ cells were plated in 96-well plates. One day later, complete medium was replaced with medium supplemented with 0.5% FBS. The day before the assay, the BrdU reagent was added overnight. On day 8 or 15, cells were then fixed and incubated with a primary detector antibody for 1 h at RT. After washing, peroxidase goat-anti-mouse antibody was added for 30 min at RT and then 3,3′,5,5′-tetramethylbenzidine (TMB) peroxidase substrate for 30 min at RT. Finally, stop solution was added and the absorbance was read to be 450 nm. Wells without BrdU reagent were used as a control for determining the background.

### 2.5. Cell Viability MTS Assay 

On day 0, 5 × 10^3^ cells were plated into 96-well plates. On day 8 or 15, a mix of MTS (Promega, Madison, WI, USA)/PMS (Sigma-Aldrich, St Louis, MI, USA) was added to wells and placed at 37 °C in an incubator (Analis, Namur, Belgium) with CO_2_ for 1 h. Absorbance was read at 490 nm. Wells without cells were used as control for determining the background. 

### 2.6. Senescence β-Galactosidase Staining

Senescence-associated β-galactosidase (SA β-gal) staining was performed, following the manufacturer’s instructions (Cell Signaling, Beverly, MA, USA). Briefly, on day 0, 15x10^3^ cells were plated into 6-well plates. On day 8 or 15, cells were rinsed with phosphate buffer saline (PBS, BioWhittaker, Walkersville, Maryland, USA) then fixed for 10 min at RT. Cells were rinsed again with PBS. A β-galactosidase staining solution (pH 6) was added to the wells. Cells were placed in a dry incubator without CO_2_ until blue color detection. Cells were visualized using an optical microscope. A total of two hundred cells was counted per condition. The percentage of SA β-gal-positive cells was then calculated.

### 2.7. Gene Expression Analysis by Real-Time Quantitative PCR

As described by Deroyer et al. [20], after the extraction of total RNAs using a Nucleospin RNA kit (Macherey-Nagel, Düren, Germany), reverse transcription was performed with a RevertAid H-Minus First-Strand cDNA Synthesis Kit (Thermo-Scientific, Pittsburgh, PA, USA) following the manufacturer’s protocol. A real-time reverse transcription quantitative PCR (RT-qPCR) was used to amplify the cDNA products with the KAPA SYBR FAST detection system (Sopachem, Nazareth, Belgium). As described by Deroyer et al., the experiments were run on a LightCycler 480 instrument (Roche Diagnostics, Mannheim, Germany) and data were analyzed using the LC480 software, release 1.5.0 SP4. For each primer, individual real-time PCR efficiency (*E* = 10(−1/slope) was calculated by generating the cDNA dilution curves. The 2^−ΔCt method was used to calculate relative gene expression between different conditions. Input amounts were normalized with the GAPDH endogenous control gene. Primers were purchased from Eurogentec (Seraing, Belgium) or Integrated DNA Technologies (Coralville, Iowa, USA). The primers used were (1) for GAPDH—forward: 5′-TGT AGT TGA GGT CAA TGA AGG G-3′; reverse: 5′-ACA TCG CTC AGA CAC CAT G-3′; (2) for p16^INK4A^—forward: 5′-GCT GCC CAA CGC ACC GAA TA-3′; reverse: 5′-ACC ACC AGC GTG TCC AGG AA-3′; (3) for p21^Cip1/Waf1^—forward: 5′-TGT CCG TCA GAA CCC ATG C-3′; reverse: 5′-AAA GTC GAA GTT CCA TCG CTC-3′. 

### 2.8. Protein Expression Analysis by Western Blotting

After determining the sample concentration using a Micro BCA Protein Assay (Thermo Fisher Scientific, Waltham, MA, USA), 20 µg of proteins were separated by SDS-PAGE and transferred to polyvinylidene difluoride membranes (EMD Millipore, Billerica, MA, USA). After blocking with milk, membranes were incubated overnight at 4 °C with primary rabbit antibody: anti-p16^INK4A^ (1:250, Abcam), anti-p21^Cip1/Waf1^ (1:1000, Cell Signaling, Beverly, MA, USA) and anti-GAPDH (1:10,000, Sigma-Aldrich). After three washes, membranes were incubated with the secondary goat anti-rabbit antibody (1:1000, Cell Signaling) at RT for 1 h. Reactions were revealed with the enhanced chemiluminescence detection reagent (ECL kit, Thermo Fisher Scientific). The intensity of each band was assessed by densitometry using the Image Studio Lite Software version 5.2 (Li-Cor Biosciences, Linkolin, NE, USA). GAPDH was used as a loading control.

### 2.9. Protein Expression Analysis by Immunofluorescence 

On day 0, 12 × 10^3^ cells were plated on coverslip into 6-well plates. On day 8, cells were fixed with 4% paraformaldehyde for 5 min at RT and blocked with PBS/bovine serum albumin/FBS/Triton solution for 30 min at RT. Cells were then incubated with a primary rabbit anti-Υh2AX antibody (1:1000; Abcam) at 4 °C overnight. After 3 washes with PBS/Triton solution, cells were incubated with the secondary goat anti-rabbit antibody (1:500, AlexaFluor 488; Abcam) for 1 h at RT. Cells were washed again with PBS and Dapi (Sigma) solution was applied on the coverslip for 30 min at RT for nucleic acid staining. Cells were mounted using Prolong Diamond Antifade Mountant (Invitrogen, Thermo Fisher Scientific). Slides were maintained at 4 °C. Images were acquired using the confocal Leica TCS SP5 (Leica Microsystems, Wetzlar, Germany) and analyzed using Fiji software.

### 2.10. Statistical Analysis

The results were expressed as mean (±SEM) or median (range). Graphs were obtained via the GraphPad Prism software (version 6.0, La Jolla, CA, USA). For unpaired samples, the Mann–Whitney test was used, while, for paired samples, a Friedman test with Dunn’s multiple comparisons or a Wilcoxon test was applied for multiple comparisons or comparisons between two groups, respectively. For all tests, *p*-values < 0.05 were considered significant (* *p* < 0.05; ** *p* < 0.01; *** *p* < 0.001; **** *p* < 0.0001). 

## 3. Results

### 3.1. Senescent p16^INK4A^-Positive Cells in OA Synovial Membrane Obtained from Biopsy

We first confirmed the presence of senescent p16^INK4A^-positive cells in the synovial membrane, obtained via biopsy in OA patients: IHC for p16^INK4A^, performed on these synovial membranes, revealed a mean (±SD) percentage of 0.73 (±0.86) p16^INK4A^-positive cells (exemplary picture of a positive area shown in Figure 1a,b). There were more p16^INK4A^-positive cells in the synovial membrane of patients at an advanced radiographic stage: median (min-max) percentage of 0.17% (0.03–0.54) for a Kellgren and Lawrence grade of 0-1, vs. 0.76% (0.13–2.93) for a Kellgren and Lawrence grade of 2-3-4, *p* = 0.01 (Figure 1c–e).

### 3.2. Influence of Prednisolone on Senescence Hallmarks in Osteoarthritic Fibroblast-like Synoviocytes (Cells Recovering after 8 Days)

Then, we analyzed in vitro the influence of the GC prednisolone on senescence parameters, in FLS obtained from OA patients. FLS were chronically stimulated (at days 1, 4 and 7) with prednisolone at three different doses (1, 5 and 10 μM) for seven days (Figure 2a). We first studied cell proliferation: after eight days of culture, the BrdU incorporation for cell proliferation was not reduced under prednisolone stimulation, with even significantly more positive cells at 10 μM (Figure 2b). The MTS analysis for cell viability showed no difference in survival (Figure 2c), leading to the same profile for the proliferation/viability ratio under prednisolone as that in Figure 2b (Figure 2d). Prednisolone stimulation did not enhance the number of β-galactosidase-positive cells (Figure 2e) at any of the three doses, with a mean percentage of 4.8% (±0.8) of positive cells in non-stimulated FLS from OA patients (representative picture in the Appendix A, Appendix A). Prednisolone did not modify DNA damage either, with no difference in γH2AX staining (Appendix A, Appendix A). Lastly, CKI mRNA and protein were analyzed: p21 mRNA levels were significantly reduced at all three doses (Figure 2f) but with no modulation for the protein (Figure 2g), while we observed an increase in p16^INK4A^ mRNA and protein for two, but not all, of the prednisolone concentrations (Figure 2h,i) (representative picture of the Western blot in Appendix A, Appendix A). In summary, we observed higher proliferation, with no accumulation of β-galactosidase-positive cells, no DNA damage and no higher expression of p21. Even in the case of higher p16^INK4A^ expression under two prednisolone dosages, we did not observe a consistent senescent phenotype. 

To ensure that there was no long-term modulation of senescence parameters after the 7-day sequence of stimulation with the GC prednisolone, we also analyzed the FLS after 15 days. FLS were stimulated over seven days with prednisolone, as we previously performed, but were analyzed after 15 days (Appendix A, Appendix A). As was similar to our results after eight days (Figure 2b–i), we observed no evident senescence hallmark apparition at day 15. BrdU incorporation (Appendix A, Appendix A) and the proliferation/viability ratio (Appendix A, Appendix A) were not reduced, with a significantly higher rate at 5 and 10 μM. In addition, there was no modulation of the number of β-galactosidase-positive cells (Appendix A, Appendix A). CKI modulation after 15 days was similar to what we observed after eight days (Figure 2f–i): there was only a reduction in p21 mRNA at 1 μM, while there was no modulation of the protein and an enhancement of p16^INK4A^ mRNA and protein at different prednisolone doses (Appendix A, Appendix A–i). 

### 3.3. Influence of Prednisolone on Senescence Induction in Osteoarthritic Fibroblast-like Synoviocytes (Cells Recovering after 8 Days)

We also wondered if GC prednisolone can modulate the induction of senescence in FLS from OA patients. We first established a model of senescence induction with irradiation: FLS were irradiated with a unique 10 Gray dose and chronically stimulated (on days 1, 4 and 7) with the GC prednisolone at three different doses (1, 5, and 10 μM) for seven days and then analyzed (Figure 3a). After irradiation, without prednisolone, we observed hallmarks of senescence, with a significant huge decrease of proliferation and proliferation/viability ratio (9.9% ± 2.1 and 15.1% ± 3.5 respectively, *p* < 0.001 and *p* < 0.05 compared to the control without irradiation) (Figure 3b,d). There were significantly more SA β-gal positive cells (Figure 3e, from 11.8 ± 2.9 to 47.6 ± 2.9, *p* < 0.001, representative picture in Appendix A, Appendix A), accumulation of γH2AX (Appendix A, Appendix A), and higher levels of p21 mRNA and protein (Figure 3f,g). Prednisolone concomitant stimulation, at any dose, did not modulate senescence implementation in FLS from OA patients: with the GC prednisolone, there was no change for the reduced proliferation, with no modification of the lower BrdU incorporation induced by irradiation (Figure 3b). Prednisolone did neither modify the accumulation of β-galactosidase positive cells (Figure 3e, representative picture in Appendix A, Appendix A) nor γH2AX staining (Supplementary materials, Appendix A). Under prednisolone stimulation, p16^INK4A^ and p21 mRNA and proteins did not differ from the irradiated controls without prednisolone stimulation (Figure 3f–i) (representative picture for Western blot in Appendix A, Appendix A).

### 3.4. Influence of Prednisolone on Senescence Hallmarks in Mesenchymal Stem Cells 

MSCs are another cell type present in the synovial membrane [21], where they exert regenerative and supportive functions to ensure joint homeostasis [22]. However, they present impaired self-renewal capacities, lost cartilage formation abilities, and exhibited reduced seno-suppressive and anabolic paracrine functions once in a senescent state [10]. Therefore, we also wanted to assess prednisolone’s influence on the senescence parameters of these cells: MSCs were stimulated with prednisolone at three different doses (1, 5, and 10 μM) for seven days and were then analyzed (Figure 4a). BrdU incorporation was not significantly modified under prednisolone stimulation, with no difference in survival or proliferation/survival ratio under prednisolone stimulation (Figure 4b–d). The number of SA β-gal-positive cells was significantly higher after prednisolone stimulation at 1, 5 and 10 µM (20.7% ± 3.2, 20.1% ± 5.5, 23.4% ± 5.0, respectively) when compared to non-stimulated MSC (5.4% ± 1.1) (representative picture in Appendix A, Appendix A). However, prednisolone did not modify DNA damage, with no difference in γH2AX immunofluorescence (Appendix A, Appendix A). CKI levels (p16^INK4A^ and p21 mRNA and protein) were not modulated as a result of seven days of prednisolone stimulation (Figure 4f–i) (representative picture for Western blot in the Appendix A, Appendix A–f).

## 4. Discussion

In the present study, we demonstrated for the first time that the GC prednisolone did not induce cellular senescence in FLS originating from OA patients. We also demonstrated that the GC prednisolone did not modify the implementation of senescence in our experimental model using OA FLS.

GC have anti-inflammatory properties and are frequently used in intra-articular injections to relieve effusion and pain in knee OA. We previously confirmed its well-known anti-inflammatory action: prednisolone effectively reduced spontaneous or TNF-α-induced IL-6, IL-8 and MMP-1 secretion in OA FLS [23]. However, GC can also cause adverse metabolic events and we demonstrated that prednisolone could induce the expression of pro-inflammatory adipokines (leptin and its receptor) [24], GILZ (a mediator of leptin expression) [25], and serum amyloid A [26] in joint cells, suggesting that GC can also be deleterious in OA. In this study, we observed no decrease in proliferation (the main characteristic of senescence), and no increase in β-galactosidase staining, DNA damage, or p21 expression in OA FLS. Only one parameter increased, that of p16^INK4A^ levels. Nevertheless, senescence is more a phenotype than a strict addition of biological criteria and p16^INK4A^ enhancement is not associated here with a reduced proliferation. The proliferation rate even increased under prednisolone, suggesting a potential protective effect. We suspected the senescence of taking time to settle, and that the eight-day timing applied for cell recovery was too short to observe senescence implementation. Therefore, experiments were repeated with cell-recovering checks after 15 days, but the results were similar. In particular, the higher proliferation rate was maintained after fifteen days. Accordingly, we kept the seven-day pattern of stimulation and did not perform any longer stimulation tests with prednisolone, which was also because the chronic injection of GC is not performed currently during OA treatment. 

Except for post-traumatic occurrences, human OA often takes place due to a progressive accumulation of lesions. Thus, we also became interested in prednisolone’s influence on experimental-induced senescence and we asked the question of whether prednisolone could modulate senescence implementation. Senescence implementation was experimentally performed via unique 10 Gy irradiation. Irradiation leads to senescence establishment, with lower cell proliferation, higher β-galactosidase staining, higher DNA damage and p21 mRNA and protein accumulation. Prednisolone neither modified experimental senescence induction in FLS nor reversed it. With no modulation of any senescence parameters, the prednisolone did not worsen the experimental senescence implementation. However, prednisolone did not allow the establishment of reverting senescence: if irradiation induced a strong reduction in the proliferation rate, concomitant prednisolone stimulation did not alleviate it.

We focused on FLS not only because GC injections are generally performed when effusion and synovial proliferation occur but also because recent publications highlighted the fact that cartilage was possibly not the only driver of senescence in the joint [11]. IHC confirmed the presence of senescent cells in our cohort of synovial biopsies; we also observed, in vitro, senescent FLS with positive β-galactosidase staining from OA synovium. MSCs are another cell type present in the synovial membrane [21]: they play a major role in joint homeostasis, as reviewed by Li et al. [27], and lose their supportive function when senescent [10]. Similar to what we observed for FLS, we found no apparent senescence phenotype induced in the presence of GC prednisolone. β-galactosidase staining was increased, but we did not observe any modulation of the other senescence parameters (proliferation rate, DNA damage, or CKI). However, caution should be exercised, because we observed a great deal of heterogeneity at the individual level. Some patients had a tendency to present a lower cell proliferation rate, although this was not statistically significant. It is of note that there was no correlation between the proliferation rate and CKI level modulation under prednisolone.

Data about GC and senescence are controversial (dose, nature of the GC, or length of the stimulation differences, etc.). In vivo data suggested that systemic GC induced senescence in the metaphysis of the long bone [28,29], with a reduction in cell proliferation. Poulsen et al. also showed an increased percentage of SA β-galactosidase-positive cells and p21 overexpression after dexamethasone in tenocytes [17]. Other in vivo studies, however, were not in accord and the researchers observed a protective effect against vascular senescence [30]. In vitro, two studies revealed higher senescence after dexamethasone stimulation in scleral fibroblast and amnion epithelial cells, respectively [31,32]. Nevertheless, other data were rather in favor of a protective role of dexamethasone against senescence by enhancing human fibroblast proliferation [33]. Prolonged exposure to dexamethasone did not accelerate telomere shortening, another senescence characteristic [34]. Carvalho et al. also provided more reassuring data regarding human fibroblasts [35]: GC were protective against senescence development, but they did not restart the proliferation of cells that were already senescent. In regard to the highly heterogeneous literature, our study focuses on knee OA where intra-articular GC injections are common but controversial. The cellular senescence phenotype was analyzed via several features including cellular proliferation, β-galactosidase staining, DNA damage, and mRNA and protein CKI expression. SASP was not analyzed, but a previous study already demonstrated that GC could suppress a selected component [36]. In addition, GC is well known to suppress pro-inflammatory cytokine secretion [23].

In our experimental conditions, we found no argument to support the theory that GC can induce cellular senescence in OA FLS. We also found no argument to incriminate GC in worsening senescence induction. In regard to the involvement of cellular senescence in the OA pathogeny and worsening with GC use in OA, this study provides reassuring in vitro data. Further research, including synovial membrane explants and in vivo animal model analyses, are needed to confirm these in vitro results.

## Figures and Tables

**Figure 1 jcm-10-05331-f001:**
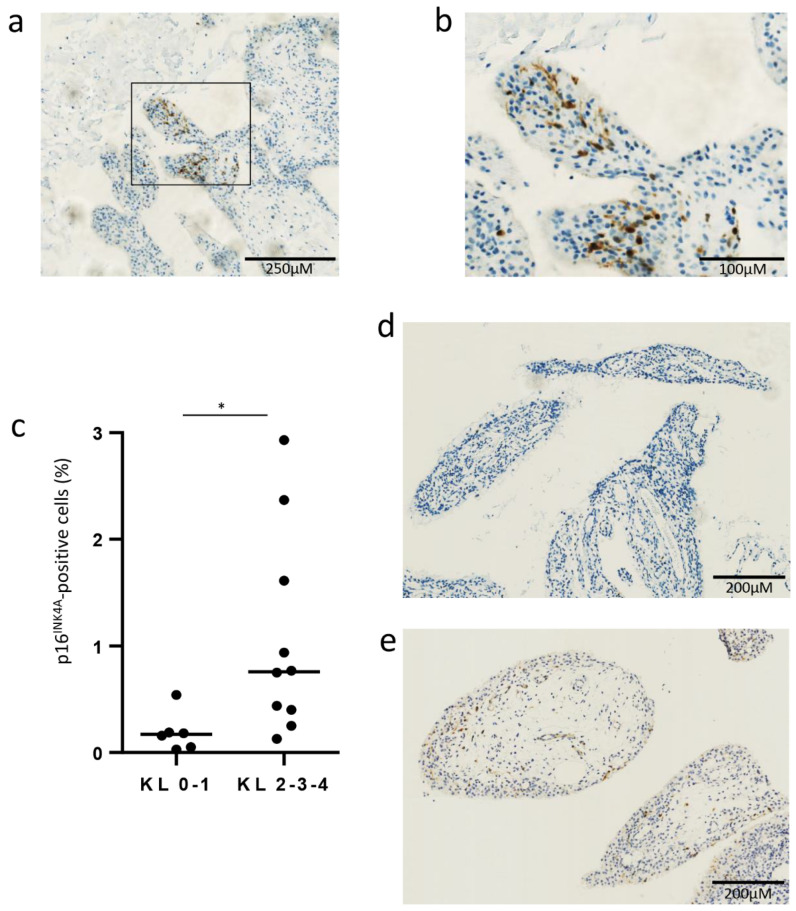
Senescent p16^INK4A^-positive cells in OA synovial membrane taken from a biopsy. (**a**,**b**) Representative pictures in a positive area of p16^INK4A^ staining in OA synovial membrane from a biopsy at magnification 4 and 10, respectively. (**c**) Number of p16^INK4A^-positive cells between patients with Kellgren and Lawrence grade 0 and 1 (KL0-1, *n* = 6) and patients with Kellgren and Lawrence grade 2, 3 and 4 (KL2-3-4, *n* = 10) (median). (**d**,**e**) Representative picture of p16^INK4A^ staining in OA synovial membrane from a patient with KL 0-1 (**d**) and from a patient with KL 2-3-4 (**e**). * *p* < 0.05.

**Figure 2 jcm-10-05331-f002:**
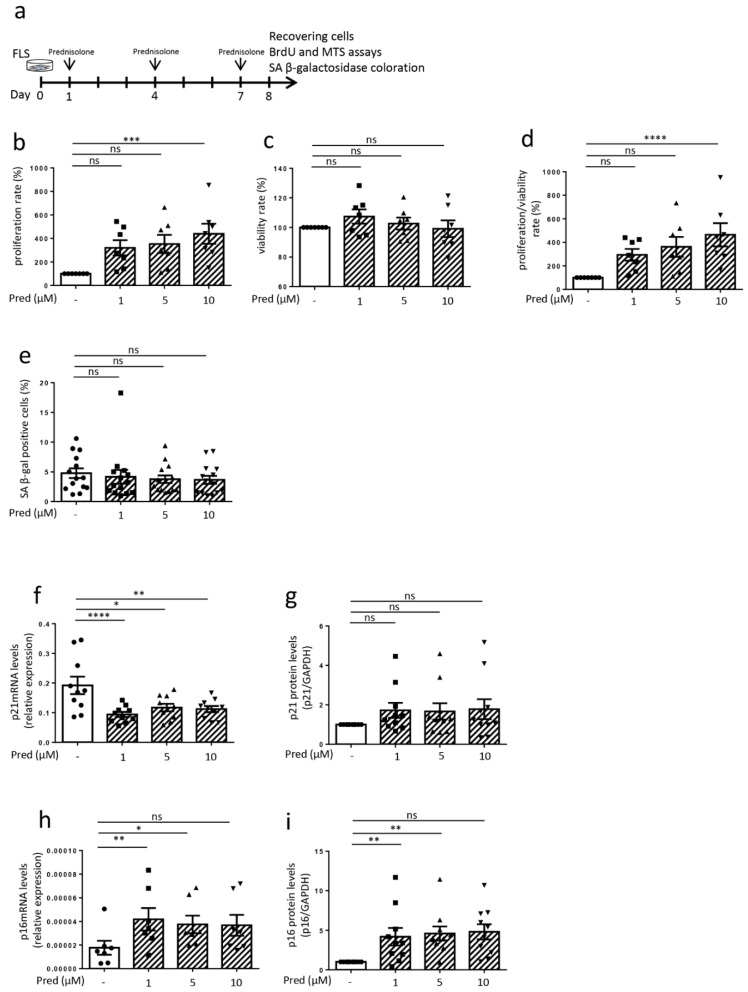
Effect of prednisolone on the hallmarks of senescence in osteoarthritic fibroblast-like synoviocytes at day 8. (**a**) Experimental design. (**b**–**d**) Proliferation rate analysis (BrdU incorporation), viability rate analysis (MTS analysis) and proliferation/viability ratio (*n* = 7). (**e**) β-galactosidase staining (*n* = 14). (**f**,**g**) p21 mRNA expression by RT-qPCR (data expressed as 2^−ΔCt, *n* = 10) and protein expression by Western blotting (levels reported on GAPDH expression, *n* = 10). (**h**,**i**) p16^INK4A^ mRNA expression by RT-qPCR (data expressed as 2^−ΔCt, *n* = 7) and protein expression by Western blotting (levels reported on GAPDH expression, *n* = 10). Results are expressed as mean ± SEM; * *p* < 0.05, ** *p* < 0.01, *** *p* < 0.001, **** *p* < 0.0001, ns = not significant.

**Figure 3 jcm-10-05331-f003:**
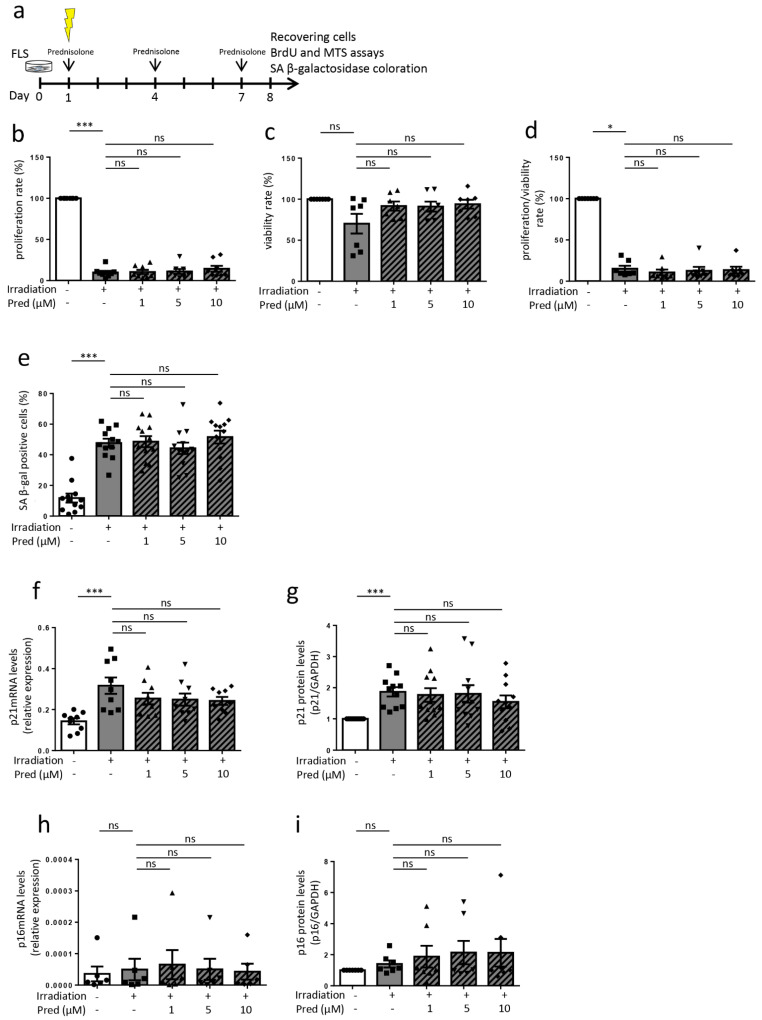
Effect of prednisolone on senescence induction in osteoarthritic fibroblast-like synoviocytes at day 8. (**a**) Experimental design. (**b**) Proliferation rate analysis (BrdU incorporation) (*n* = 8). (**c**) Viability rate analysis (MTS analysis) (*n* = 7). (**d**) Proliferation/viability ratio (*n* = 7). (**e**) β-galactosidase staining (*n* = 12). (**f**,**g**) p21 mRNA expression by RT-qPCR (data expressed as 2^−ΔCt, *n* = 9) and protein expression by Western blotting (levels reported on GAPDH expression, *n* = 11). (**h**,**i**) p16^INK4A^ mRNA expression by RT-qPCR (data expressed as 2^−ΔCt, *n* = 6) and protein expression by Western blotting (levels reported on GAPDH expression, *n* = 7). Results are expressed as mean ± SEM; * *p* < 0.05, *** *p* < 0.001, ns = not significant.

**Figure 4 jcm-10-05331-f004:**
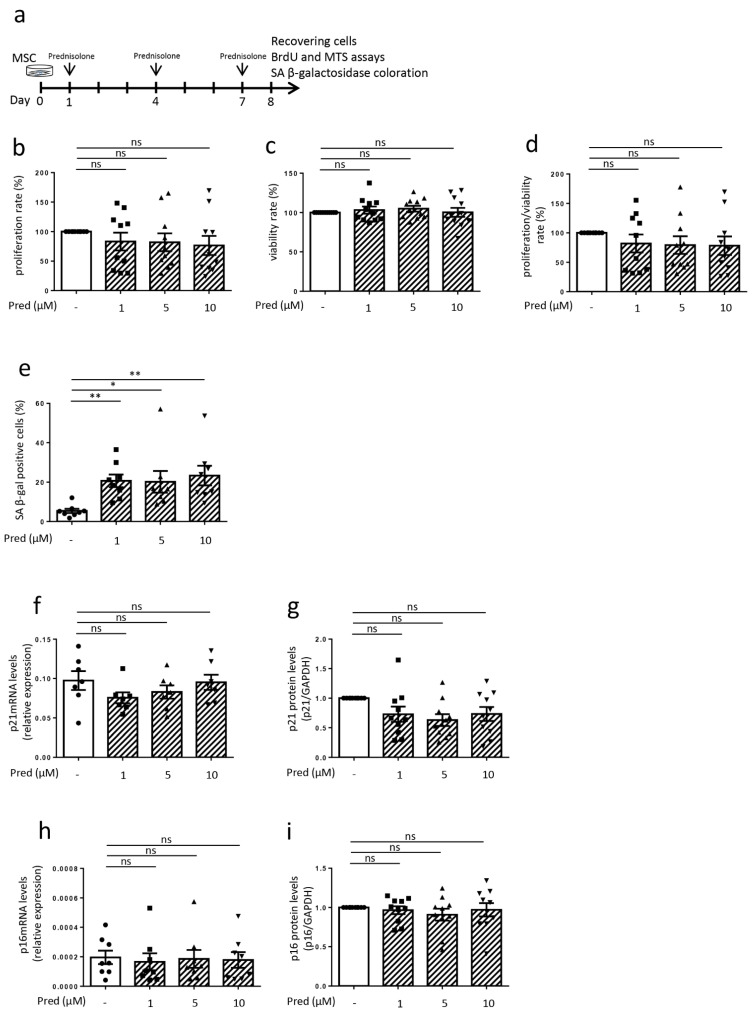
Effect of prednisolone on hallmarks of senescence in mesenchymal stem cells at day 8. (**a**) Experimental design. (**b**) Proliferation rate analysis (BrdU incorporation) (*n* = 10). (**c**) Viability rate analysis (MTS analysis) (*n* = 11). (**d**) Proliferation/viability ratio (*n* = 10). (**e**) β-galactosidase staining (*n* = 8). (**f**,**g**) p21 mRNA expression by RT-qPCR (data expressed as 2^−ΔCt, *n* = 7) and protein expression by Western blotting (levels reported on GAPDH expression, *n* = 10). (**h**,**i**) p16^INK4A^ mRNA expression by RT-qPCR (data expressed as 2^−ΔCt, *n* = 8) and protein expression by Western blotting (levels reported on GAPDH expression, *n* = 10). Results are expressed as mean ± SEM; * *p* < 0.05, ** *p* < 0.01, ns = not significant.

## Data Availability

The data presented in this study are available on reasonable request from the corresponding author.

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
