# Peer review of "Influence of Glucocorticoids on Cellular Senescence Hallmarks in Osteoarthritic Fibroblast-like Synoviocytes"

_jcm, 2021, doi:10.3390/jcm10225331_

Round 1

Reviewer 1 Report

This study aims to investigate the potential influence of the glucocorticoid prednisolone on the development of cellular senescence within the synovial membrane of human osteoarthritis patients.  While the importance of understanding the possibilities of adverse effects of current treatments is clear, the experiments conducted within this manuscript are very preliminary and lack some controls.  Ideally this research questions should be addressed using in vivo models and human tissue explants rather than purely in in vitro cultured cells outside of the articular joint.  The presentation of negative findings is commended, however the experimental design should be modified and additional controls and functional readouts included in order to support any conclusion.

  • The complete staining protocol for p16INK4a is missing.This should include secondary antibodies when applicable, detection methods and counterstains.
  • What are the KL scores of the patient samples used for cell culture experiments? The range of the samples used here may make a contribution to the large variability seen in the data sets.
  • Figure 1 should include representative images of both low and high KL score samples to demonstrate the difference seen in the quantification.The staining appears to be very localised, what is specific about these areas?  Within image 1b alone, there appears to be a lot more than 2% positive cells, how is the quantification calculated?
  • The figure legend in figure 1 states that n = 16, however only 16 data points are plotted in total within the 2 groups together.Please correct to show the n numbers referring to the group sizes rather than the total of all samples.
  • Why were these doses of prednisolone of used?In order to conclude that IA injection of prednisolone does not cause cellular senescence in the synovial membrane, the concentration and duration that the synovium is exposed to physiologically during treatment should be used.
  • Experiments involving western blotting are missing representative gels, b-galactosidase stainings are missing representative images and positive controls.
  • The experiments examining the effects of prednisolone on the development of senescence following irradiation do not seem clinically relevant.What evidence is there to suggest this type of irradiation leads to a rate of senescence equivalent to that found in the synovium of OA patients, without additional effects?
  • When studying the effects of prednisolone on MSCs present in the synovial membrane, why do authors instead isolate MSCs from the bone marrow?It is also possible to isolate and culture human synovial membrane MSCs, which would be more suitable for testing this hypothesis.
  • It is not clear what the authors refer to within the discussion lines 319-320 regarding the ‘spreading’ of senescence.The role of chondrocyte senescence in OA has been well demonstrated (e.g. Jeon et al. 2017 Nature Medicine PMID: 28436958) therefore it is misleading to suggested that senescence within the cartilage does not play a role in OA progression.  Senescent cells within the synovial membrane may well secrete a SASP, but this is unlikely to be the only effector of ultimate joint failure. 

Reviewer 2 Report

Summary:

Glucocorticoids (GC) are often and controversial used in knee osteoarthritis. Positive and negative effects are seen and discussed. The authors want to analyse the effect of glucocorticoids on cell senescence. Therefore, they did an in-vitro analysis on fibroblast-like synoviocytes (FLS) and mesenchymal stem cells.

In addition, they perform a senescence process by irradiation for FLS and analyse the ability of GC to stop this process. Main result: Glucocorticoids did not induce a senescence phenotype in vitro for both, FLS and MSc. In contrast, proliferation rate was increased in FLS. However, a started senescence process was not reversible by GC.

M&M:

Fine for me, cell sources and analysis methods are explained in detail. However, they used 24 specimens for FLS cultures. Did they 24 parallel analyses with n=3, 5 or more for any FLS culture? (16 samples instead of 24 for Figure 1)? Because in the legends, I can find n=7, n=10, n=14, n=16. What does it mean?

Results and diagrams:

Results are clear and interesting, no concerns. Well demonstrated by diagrams.

Discussion/Conclusion:

Results were discussed with the existing literature. Limitation as in vitro study is mentioned “reassuring in vitro data”. However, it would be nice to understand what is the next step? In vivo model? What is the clinical relevance? Only presumptions? (Because of in vitro data, or is there more?)
